# ADAM: A Diverse Archive of Mankind for Evaluating and Enhancing LLMs in Biographical Reasoning

## Abstract

We introduce **ADAM** (A Diverse Archive of Mankind), a framework for evaluating and improving multimodal large language models (MLLMs) in biographical reasoning. To the best of our knowledge, this is the first work to systematically examine LLM capabilities in biography, a critical yet underexplored dimension of factual knowledge. At its core, **AdamDB** is a multilingual and multimodal dataset covering over 4 million individuals across geography, time, and profession, while **AdamBench** provides cognitively structured evaluations based on Bloom's taxonomy, spanning six reasoning levels in both English and native languages. To address hallucinations, particularly for lesser-known individuals, we propose **AdamRAG**, a retrieval-augmented generation system tailored to biographical contexts. Experiments show that AdamRAG substantially improves open-source models and modestly benefits closed-source ones, with the largest gains on lower-order reasoning. Popularity strongly mediates accuracy, and multimodal input via face images offers smaller, less consistent improvements than retrieval. ADAM establishes the first benchmark and framework for cognitively, culturally, and multimodally grounded biographical evaluation, advancing the development of multilingual, accurate, and hallucination-resistant MLLMs.

## 1 Introduction

The proliferation of Large Language Models (LLMs) has revolutionized information access, yet their application to biographical content reveals critical vulnerabilities. This domain demands absolute factual accuracy, but is plagued by LLM "hallucinations" (the generation of fabricated or incorrect facts). This problem is compounded by the poor performance of even advanced vision-language models like Qwen-VL Bai et al. (2025) and Gemma Team et al. (2025) on multimodal biographical reasoning tasks, the lack of popularity-aware systems that can distinguish between a global icon and a regional figure, and a persistent English-centric bias. Consequently, retrieving reliable biographical information across the world's languages and cultures remains a significant challenge, creating a gap where misinformation can flourish.

Existing resources are ill-equipped to solve this. Historical and semantic biographical systems such as BiographySampo (Hyvönen et al., 2019), while valuable, are typically monolingual, manually curated, and limited in scale. This reliance on manual curation results in incomplete and static collections that lack linguistic and cultural diversity. While the field has evolved from traditional information retrieval to more sophisticated retrieval-augmented generation (RAG) and graph search methods, no system has been specifically engineered to handle the unique complexities of biography. There is currently no robust, multilingual, popularity-aware, and multimodal RAG system capable of navigating the vast and varied landscape of human life stories, leaving significant temporal, cultural, and linguistic gaps in our digital knowledge.

To address these gaps, we introduce **ADAM** (A Diverse Archive of Mankind), the first comprehensive, retrieval-augmented framework specifically designed for biographical reasoning across global languages. At its core, **AdamDB** contains over four million structured biographical records spanning nearly 600 native languages, automatically constructed via a WikiDB-to-RAG pipeline. To overcome linguistic barriers, ADAM integrates cross-language entity linking and multilingual se-

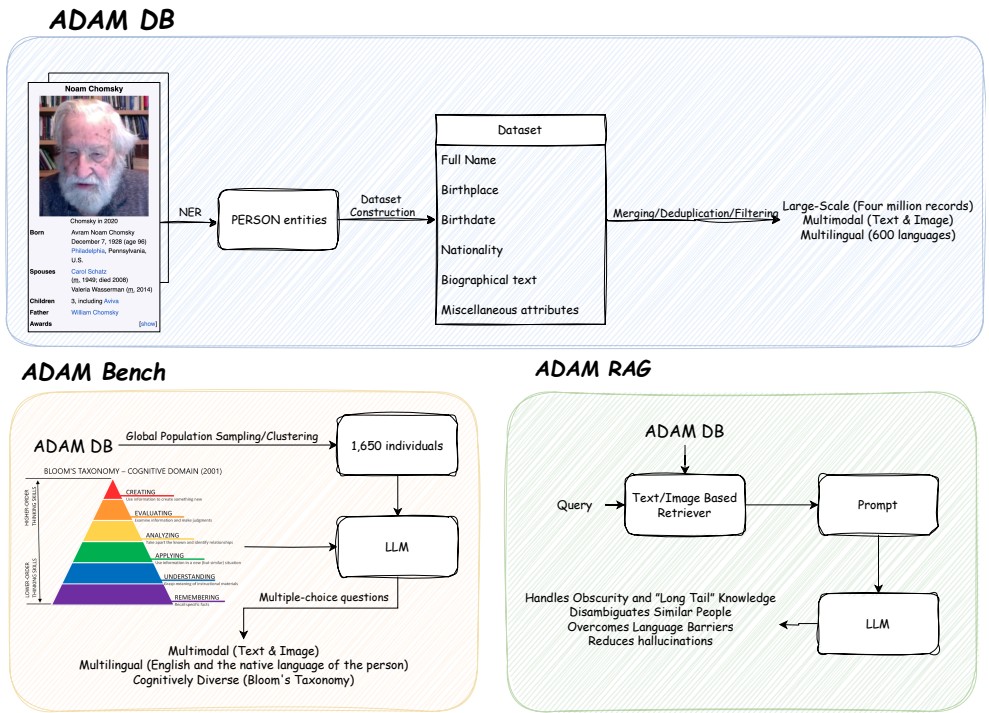

Figure 1: Overview of the **ADAM** framework. **AdamDB** builds a large-scale, multilingual, and multimodal biographical knowledge base. **AdamBench** enables cognitively diverse and cross-linguistic evaluation through Bloom's taxonomy, spanning all six reasoning levels. **AdamRAG** integrates retrieval-augmented generation to strengthen factual grounding, improve accuracy, and mitigate hallucinations in biographical reasoning.

mantic search. To incorporate awareness of subject popularity, **AdamRAG** introduces popularity-weighted retrieval using Wikipedia engagement metrics, enabling adaptive knowledge access. Finally, to move beyond fact-checking, **AdamBench** provides a cognitively rigorous evaluation suite grounded in Bloom's taxonomy, supporting systematic assessment of biographical reasoning from factual recall to creative synthesis.

**Contributions:** The main contributions of this work are **(i) ADAM Framework:** We present the first retrieval-augmented framework for biographical reasoning, integrating multilingual retrieval, popularity awareness, and cognitive evaluation in a unified system. **(ii) AdamDB:** A large-scale, multilingual, and multimodal biographical knowledge base covering over four million individuals across time, geography, profession, and nearly 600 languages. **(iii) AdamBench:** A biographical reasoning benchmark grounded in Bloom's taxonomy, providing a structured cognitive evaluation across six hierarchical levels in both English and subjects' native languages. **(iv) AdamRAG:** A retrieval-augmented generation system tailored for biography, incorporating cross-lingual retrieval and popularity-weighted search, significantly reducing hallucinations and improving factual grounding, especially for lesser-known individuals. **(v) Comprehensive Evaluation:** We provide the first systematic analysis of LLMs and MLLMs on biographical reasoning tasks, highlighting gaps in cross-linguistic generalization, cognitive depth, popularity bias, and multimodal grounding. By integrating retrieval with cognitive and multilingual benchmarks, ADAM offers the community a robust framework for developing more accurate, culturally grounded, and hallucination-resistant LLMs.

## 2 RELATED WORKS

Biographical datasets consist of structured or unstructured narratives that capture key details of individuals' lives, including their personal history, career developments, relationships, and significant life events. They serve as crucial resources for a variety of natural language processing tasks such

as biography summarization, named entity recognition, information extraction, and text generation. These datasets range widely in size and focus from large-scale collections with millions of records to more specialized corpora centered on particular professions, time periods, or population groups. Typical sources include encyclopedias, web-scraped data, news articles, social media content, and synthetic profiles, each offering varying levels of accuracy, coverage, and structure.

A wide range of datasets have been developed for biographical text generation, information extraction, and related NLP tasks. WikiBio Dataset Lebret et al. (2016) consists of 728,321 English Wikipedia biography entries linked with infoboxes, focusing on generating the first sentence of each article. Bamman & Smith (2014) is derived from the 2014 Wikipedia dump, comprising 927,403 entries with structured person data metadata. The filtered version contains 242,970 biographies of individuals born after 1800 with at least five documented life events. BigWikiBio Ambavi et al. (2020) expands upon WikiBio, offering nearly 6 million biography articles scraped from English Wikipedia. BIOS Dataset De-Arteaga et al. (2019) offers 400,000 concise biographies across 28 occupations, extracted from Common Crawl using occupation-based filtering aligned with the BLS taxonomy. It processes WET files from 16 crawls conducted between 2014 and 2018. SynthBio Yuan et al. (2021) is a synthetic benchmark for WikiBio, consisting of 2,249 fictional infoboxes paired with 4,692 generated biographies, each infobox mapped to an average of 2.1 biographies. Pantheon 1.0 Yu et al. (2016) offers manually curated biographies of 11,341 globally notable individuals, enriched with occupation categories and popularity metrics such as the Historical Popularity Index. BiographySampo (Finnish National Biography) Hyvönen et al. (2019) presents biographical data, including structured metadata such as occupations, birth/death dates, and author demographics. EventKG Gottschalk & Demidova (2019) includes person-centric timelines with annotated events for training and evaluation, covering professions such as politics, music, and sports. Biographical Relation Extraction Dataset Plum et al. (2022) enables relation extraction across ten predefined categories by aligning Wikipedia sentences with data from Pantheon and Wikidata, including a manually verified evaluation set. As shown in Table 1, ADAM significantly outperforms previous datasets in terms of records, supported languages, and country coverage.

Table 1: Comparison of ADAM with previous datasets, showing its superiority in number of records, supported languages, and global coverage.

| System | Records | Languages | Countries |
| --- | --- | --- | --- |
| BiographySampo | 13,100 | 1 (Finnish) | 1 |
| BiographyNet | 125,000 | 1 (Dutch) | 1 |
| Networked Pantheon | 11,341 | Limited | Global |
| EventKG+BT | - | Limited | Limited |
| **ADAM** | **4,016,647** | **595** | **global** |

## 3 APPROACH

### 3.1 OVERVIEW

This work introduces a comprehensive framework designed to enhance and evaluate the biographical reasoning capabilities of Large Language Models (LLMs). The foundation of this framework is AdamDB, a large-scale, multilingual, and multimodal database containing structured information and embeddings for approximately 4 million individuals, created to serve as a factual grounding resource that combats AI hallucinations and addresses coverage gaps in existing datasets. Building on this resource, we developed AdamBench, a novel evaluation benchmark featuring multilingual and multimodal questions structured according to Bloom's Taxonomy to assess a spectrum of cognitive skills, from factual recall to complex reasoning. To operationalize this data, we present AdamRAG, a Retrieval-Augmented Generation system that queries AdamDB to provide LLMs with verified contextual information, significantly improving response accuracy, especially for less-prominent individuals, and enabling robust disambiguation through sophisticated text and image-based retrieval pipelines. Figure 1 provides a holistic overview of the ADAM framework, from algorithmic data selection and sampling to benchmark generation and retrieval-augmented evaluation.

Table 2: Accuracy of leading open-source and closed-source models (🔒) on biographical reasoning tasks, organized along the six cognitive levels of Bloom's taxonomy ("Remembering" to "Creating"). For each skill, performance is reported in English ("En") and in the subject's original language ("Org"), determined by city of birth). A varying number of stars (★) indicates the popularity level of the individual. Rows specify model configurations, distinguishing multimodal input (✓ Face Image) and whether answers were generated in a two-stage retrieval setting (✓ RAG) or via zero-shot prompting.

| Model | Language | Image | RAG | (i) Remembering ★ | ★★ | ★★★ | (ii) Understanding ★ | ★★ | ★★★ | (iii) Applying ★ | ★★ | ★★★ | (iv) Analyzing ★ | ★★ | ★★★ | (v) Evaluating ★ | ★★ | ★★★ | (vi) Creating ★ | ★★ | ★★★ |
|---|---|---|---|---|---|---|---|---|---|---|---|---|---|---|---|---|---|---|---|---|---|
| Gemini Flash 2.5 | En | ✗ | ✗ | 73.2 | 82.4 | 89.5 | 87.5 | 93.1 | 97.0 | 89.4 | 95.1 | 97.2 | 87.0 | 92.8 | 91.1 | 90.7 | 96.3 | 96.2 | 90.7 | 94.7 | 96.2 |
| Gemini Flash 2.5 | Org | ✗ | ✗ | 86.5 | 92.3 | 93.4 | 89.2 | 94.2 | 96.6 | 89.2 | 93.1 | 94.4 | 85.9 | 91.0 | 89.7 | 87.3 | 93.8 | 94.2 | 88.5 | 92.3 | 94.4 |
| GPT-4 | En | ✗ | ✗ | 63.5 | 72.2 | 78.3 | 88.9 | 91.2 | 92.5 | 83.4 | 87.3 | 88.1 | 85.6 | 87.0 | 88.9 | 91.1 | 90.3 | 86.1 | 91.3 | 92.3 | 94.0 |
| GPT-4 | Org | ✗ | ✗ | 63.1 | 68.0 | 77.7 | 87.8 | 91.2 | 92.3 | 82.9 | 86.6 | 88.1 | 82.0 | 88.6 | 88.9 | 89.2 | 91.0 | 84.1 | 92.3 | 90.9 | 93.4 |
| Gemma3-12b-it | En | ✗ | ✗ | 19.0 | 22.9 | 45.7 | 57.0 | 59.5 | 69.4 | 45.6 | 53.2 | 63.8 | 46.8 | 45.8 | 51.5 | 37.2 | 40.7 | 51.7 | 25.4 | 35.0 | 53.9 |
| Gemma3-12b-it | Org | ✗ | ✗ | 20.8 | 25.2 | 39.4 | 46.7 | 48.6 | 52.7 | 37.9 | 44.5 | 51.5 | 36.9 | 37.2 | 40.6 | 28.8 | 34.3 | 44.1 | 20.6 | 28.9 | 39.0 |
| Qwen2.5-7b | En | ✗ | ✗ | 19.7 | 20.8 | 26.4 | 47.7 | 47.7 | 53.7 | 38.9 | 43.0 | 46.7 | 36.4 | 35.2 | 34.6 | 31.2 | 32.6 | 39.4 | 18.5 | 23.6 | 36.0 |
| Qwen2.5-7b | Org | ✗ | ✗ | 18.9 | 21.5 | 25.1 | 35.0 | 35.4 | 33.8 | 32.1 | 28.5 | 31.0 | 25.0 | 24.1 | 25.5 | 20.2 | 24.8 | 25.7 | 13.7 | 18.3 | 22.5 |
| Gemini Flash 2.5 | En | ✗ | ✓ | 98.3 | 99.3 | 98.2 | 96.6 | 97.5 | 96.4 | 95.2 | 94.4 | 91.1 | 93.8 | 93.3 | 88.5 | 93.1 | 93.0 | 91.3 | 94.9 | 93.0 | 90.9 |
| Gemini Flash 2.5 | Org | ✗ | ✓ | 98.8 | 98.6 | 98.6 | 97.8 | 97.5 | 99.2 | 97.1 | 95.6 | 93.4 | 93.7 | 94.7 | 92.6 | 95.4 | 96.3 | 93.6 | 96.7 | 96.1 | 94.4 |
| GPT-4 | En | ✗ | ✓ | 95.5 | 95.6 | 95.8 | 96.6 | 96.7 | 96.4 | 89.9 | 90.7 | 91.5 | 94.3 | 94.9 | 96.4 | 92.3 | 92.1 | 88.9 | 97.4 | 96.8 | 95.4 |
| GPT-4 | Org | ✗ | ✓ | 96.2 | 96.3 | 96.2 | 97.8 | 97.7 | 97.4 | 90.9 | 91.7 | 90.1 | 92.6 | 94.5 | 94.6 | 92.3 | 92.3 | 88.7 | 97.1 | 96.8 | 96.8 |
| Gemma3-12b-it | En | ✗ | ✓ | 96.9 | 95.8 | 85.7 | 86.8 | 85.6 | 74.6 | 84.2 | 78.5 | 68.0 | 74.1 | 70.8 | 63.6 | 79.6 | 76.9 | 71.0 | 77.0 | 73.8 | 66.2 |
| Gemma3-12b-it | Org | ✗ | ✓ | 94.7 | 94.7 | 82.1 | 65.7 | 67.1 | 56.1 | 64.0 | 62.0 | 52.5 | 56.1 | 51.2 | 46.3 | 55.2 | 54.4 | 47.7 | 55.8 | 52.6 | 43.1 |
| Qwen2.5-7b | En | ✗ | ✓ | 96.6 | 97.2 | 95.8 | 71.9 | 74.7 | 68.2 | 66.9 | 68.5 | 63.8 | 56.8 | 53.4 | 51.3 | 59.2 | 60.0 | 58.1 | 58.2 | 61.8 | 60.0 |
| Qwen2.5-7b | Org | ✗ | ✓ | 86.5 | 87.7 | 81.7 | 49.9 | 48.9 | 42.0 | 45.8 | 41.4 | 40.4 | 38.4 | 34.5 | 32.8 | 34.5 | 35.6 | 34.8 | 34.3 | 35.0 | 29.6 |
| Gemini Flash 2.5 | En | ✓ | ✓ | 65.7 | 73.8 | 84.9 | 88.0 | 91.0 | 95.8 | 87.8 | 92.6 | 96.6 | 83.4 | 88.4 | 91.1 | 87.8 | 95.6 | 94.8 | 91.8 | 92.3 | 95.2 |
| Gemini Flash 2.5 | Org | ✓ | ✓ | 65.9 | 72.5 | 85.9 | 86.3 | 89.6 | 95.2 | 88.0 | 93.0 | 95.0 | 83.4 | 87.0 | 91.3 | 88.2 | 94.0 | 93.2 | 89.5 | 92.3 | 95.0 |
| GPT-4 | En | ✓ | ✓ | 24.9 | 23.4 | 31.0 | 78.9 | 82.8 | 86.9 | 83.7 | 87.7 | 90.7 | 77.4 | 82.8 | 83.9 | 88.2 | 90.7 | 88.9 | 88.2 | 87.9 | 89.9 |
| GPT-4 | Org | ✓ | ✓ | 24.9 | 24.1 | 34.2 | 81.7 | 83.5 | 88.1 | 80.1 | 86.3 | 87.7 | 74.1 | 76.8 | 78.7 | 85.8 | 86.8 | 86.9 | 85.1 | 84.0 | 87.9 |
| Gemma3-12b-it | En | ✓ | ✓ | 40.3 | 42.3 | 48.9 | 68.1 | 71.1 | 76.5 | 73.1 | 79.9 | 84.1 | 71.7 | 75.7 | 79.5 | 69.0 | 80.6 | 81.1 | 68.1 | 74.3 | 79.5 |
| Gemma3-12b-it | Org | ✓ | ✓ | 38.5 | 42.1 | 48.1 | 67.5 | 70.0 | 76.2 | 72.0 | 79.8 | 84.0 | 71.7 | 75.6 | 79.2 | 68.2 | 80.5 | 80.2 | 67.5 | 74.3 | 78.3 |
| Qwen2.5-7b | En | ✓ | ✓ | 34.0 | 38.6 | 40.2 | 67.9 | 71.3 | 75.9 | 75.3 | 82.4 | 83.5 | 70.3 | 75.7 | 77.5 | 71.4 | 81.0 | 84.9 | 73.1 | 78.9 | 79.7 |
| Qwen2.5-7b | Org | ✓ | ✓ | 32.8 | 37.3 | 37.2 | 65.2 | 67.4 | 73.4 | 65.9 | 78.7 | 80.9 | 60.4 | 65.0 | 69.2 | 64.7 | 73.6 | 75.8 | 65.9 | 68.0 | 68.6 |

## 3.2 DATASET

**AdamDB** AdamDB forms the core of the ADAM framework and is specifically designed for biographical information at scale. Its key characteristics are: **(i) Large-Scale:** covering over four million individuals with both structured and unstructured records. **(ii) Multimodal:** integrating textual biographies, names, dates, and references to images. **(iii) Multilingual:** spanning nearly 600 languages, moving beyond the English-centric scope of prior datasets.

AdamDB addresses several critical limitations in existing resources: **(i) Reducing Hallucinations:** by grounding models in structured data through AdamRAG, lowering the risk of fabricated details. **(ii) Expanding Coverage:** by including diverse professions, geographies, and time periods, overcoming the narrow focus of manually curated, English-dominated databases. **(iii) Enabling Cognitive Evaluation:** by supporting AdamBench, which generates biographical questions across Bloom's taxonomy for nuanced reasoning assessment. **(iv) Improving Performance:** by significantly enhancing accuracy, particularly for less popular individuals and non-English settings, when coupled with retrieval-augmented generation.

**Data Pipeline** We begin with the WikiDBS dataset, a large collection of relational tables. A two-stage filtering process isolates human-centric content: first, foreign key columns are heuristically matched with person-related patterns (e.g., "surname"); second, Named Entity Recognition (NER) is applied to retain only columns dominated by PERSON entities. Structured biographical records are then extracted row by row, merged via name-based mappings, and validated with NER.

To ensure uniqueness, we align records with Wikidata Q-Ids and consolidate duplicates by selecting modal values for biography, nationality, birth date, and birthplace. Translations of names across languages are retrieved from Wikidata, yielding a multilingual knowledge base. We retain only individuals with non-null entries for biography, birth date, nationality, and birthplace. To quantify popularity, annual page views of each subject's English Wikipedia entry (2024) are recorded, discarding entries with zero views. A minimum of 10 individuals per country or territory is enforced to ensure global representation.

**Dataset Statistics** AdamDB ultimately contains approximately four million unique individuals. While English has the highest coverage, other major languages remain well represented. Geographic representation spans all continents and over 200 countries. Summary statistics on distribution by continent, country, and language coverage are provided in Appendix 2, Appendix 3, and Appendix 3.

**AdamBench**
AdamBench is a specialized evaluation benchmark created as part of the ADAM framework. It is a large suite of multiple-choice questions designed specifically to test how well Large Language Models (LLMs) can reason about biographical information. They are systematically generated using the data from AdamDB. They are designed to be: (1) **Multilingual**: Questions are written in both English and the native language of the person in question. (2) **Multimodal**: Some questions incorporate images, forcing the AI to connect textual information with visual data. (3) **Cognitively Diverse**: The questions are grounded in a framework called Bloom's Taxonomy to test different levels of thinking.

Having a benchmark like AdamBench is critical for several reasons, especially in the domain of biography: (1) **Measures Beyond Factual Recall**: It's easy for an LLM to repeat a birth date it found online. It's much harder for it to understand the significance of a person's life, compare their work to a contemporary, or evaluate their impact. A good benchmark tests this deeper understanding, not just rote memorization. (2) **Exposes Hallucinations and Bias**: Biographies are a prime area for LLM "hallucinations". A standardized benchmark can systematically probe for these errors and reveal biases in the model. (3) **Drives Progress**: By providing a consistent and challenging test, AdamBench allows researchers to compare different models, measure the impact of new techniques, and identify specific weaknesses that need to be addressed in future AI development.

Bloom's Taxonomy is a hierarchical model used in education to classify different levels of intellectual behavior and thinking. It's a framework that moves from simple information recall to more complex, abstract thought processes. The levels are: (1) **Remembering**: Recalling facts and basic concepts. (e.g., "When was Albert Einstein born?") (2) **Understanding**: Explaining ideas or concepts. (e.g., "Explain the basic principle of Einstein's theory of relativity.") (3) **Applying**: Using

information in new situations. (e.g., "How would Einstein's work apply to GPS technology?") (4) **Analyzing**: Drawing connections among ideas; comparing and contrasting. (e.g., "Compare the contributions of Albert Einstein and Isaac Newton to physics.") (5) **Evaluating**: Justifying a stand or decision; critiquing. (e.g., "Evaluate the ethical implications of the research Einstein's work enabled.") (6) **Creating**: Producing new or original work. (e.g., "Propose a hypothetical dialogue between Einstein and a modern physicist about quantum computing.")

By building its questions around Bloom's Taxonomy, AdamBench becomes a much more powerful and insightful evaluation tool. Instead of just asking simple "Remembering" questions, AdamBench tests the full cognitive spectrum. This allows you to see if an LLM can truly reason about a person's life. This structured approach allows the ADAM framework to pinpoint exactly where a model excels or fails. The paper's results show this clearly, for instance, noting that RAG helps most on the "lower levels of cognition" (like Remembering and Understanding). Without the Bloom's framework, it would be impossible to arrive at such a specific and useful conclusion. To construct a diverse benchmark, we first implement a proportional sampling strategy based on global population data. Individuals are grouped by country, and a base cluster count $k$ is calculated for each using the formula:

$$k = \lceil (\text{country population proportion} \times 5) + 0.01 \rceil \tag{1}$$

This ensures that every country is represented by at least one cluster. Within each country, individuals are then stratified into three popularity tiers: high (the top $5k$ individuals), medium (the top 75% excluding the high-tier), and low (the bottom 25%). $k$-means clustering is performed independently on each tier, using a feature vector composed of the individual's birth date and biography. To create this vector, birth dates are quantized to the nearest 50 years, and biographies are encoded into normalized vectors using a BERT model after removing explicit mentions of age or nationality. The birth date feature is weighted to balance its influence with the biography embedding. From the resulting clusters, we select the individual closest to the centroid for the high and medium tiers, and the individual with the highest annual visits for the low tier. This procedure yields a final set of approximately 1,650 individuals, ensuring diversity across nationality, historical period, profession, and notability.

For each of the selected individuals, we compile their names in multiple languages along with their Wikipedia summary. This consolidated information is then supplied to a Large Language Model (LLM). The LLM is prompted to perform two tasks: first, to synthesize a concise biography, and second, to generate a set of multimodal and multilingual questions based on Bloom's Taxonomy. These questions are formulated in both English and the individual's native language to create the final benchmark dataset.

### 3.3 ADAMRAG

**Motivation** AdamRAG is the retrieval-augmented generation (RAG) module of ADAM, designed to ground LLM outputs in factual biographical knowledge. Unlike open-ended text generation, biography requires accuracy: there is a correct answer to whether an individual was born in a given year or pursued a particular profession. Standard LLMs often hallucinate when information is sparse, ambiguous, or non-English. AdamRAG addresses these challenges by retrieving structured facts from AdamDB before generation. This is particularly valuable for lesser-known individuals, for disambiguating people with similar names, and for linking multilingual aliases.

**Mechanism** For a user query (e.g., "What were the major challenges Marie Curie faced in her early career?"), AdamRAG: **(i)** retrieves relevant entries from AdamDB; **(ii)** augments the query with the retrieved context; **(iii)** forwards the enriched prompt to the LLM. This pipeline ensures the answer is anchored in factual context rather than relying solely on pretraining knowledge.

**Retrieval Pipeline** We designed a multi-stage pipeline to handle both text-based and image-based queries:

*Text-based disambiguation.* The system first attempts exact matches in AdamDB. If ambiguous, it uses Language-Agnostic BERT Sentence Embeddings (LaBSE) (Feng et al., 2022) to retrieve semantically similar candidates. Results are sequentially filtered by nationality (normalized to modern countries) and birth date (±20 years). The final candidate is selected via cosine similarity between biography embeddings and the query context.

*Image-based retrieval.* For face queries, embeddings are extracted and used to retrieve the top-100 similar entries in AdamDB. These are filtered by nationality and birth date, yielding up to five candidates. To improve coverage, we crawled two verified images per individual when Wikipedia photos were absent, ensuring quality and uniqueness.

**System Evaluation** The augmented queries, containing retrieved context, are then passed to an LLM. We evaluate this system across open-source and proprietary models, under multilingual and multimodal conditions, using few-shot prompting. Analysis is stratified by Bloom's taxonomy level, subject popularity, language, and input modality. Ablation studies compare AdamRAG against zero-shot prompting. Results show AdamRAG consistently improves factual accuracy, reduces hallucinations, and narrows performance gaps between open- and closed-source models, with the greatest benefits for lesser-known individuals and lower-order reasoning tasks.

### 3.4 EVALUATION FRAMEWORKS AND METRICS

**Benchmarking Instrument** Our main evaluation tool is **AdamBench**, a benchmark of multiple-choice questions designed to probe biographical reasoning across Bloom's taxonomy. Questions span all six cognitive levels, from factual recall to creative synthesis, and are authored in both English and subjects' native languages to capture cross-linguistic generalization.

**Metric** We report **accuracy** as the principal evaluation metric. In a multiple-choice setting, accuracy provides a direct and interpretable measure of a model's ability to select the correct answer among distractors, reflecting its factual grounding, comprehension, and reasoning capacity.

**Frameworks** To ensure reproducibility and comparability, we integrate two complementary evaluation frameworks: **(i) EleutherAI Language Model Evaluation Harness (lm-eval-harness):** a widely adopted tool for standardized evaluation, enabling consistent testing across open-source and proprietary models. **(ii) Khayyam Challenge** (Ghahroodi et al., 2024; 2025): a platform specifically designed for multilingual and multimodal benchmarks, ensuring that AdamBench's diverse question types are presented and evaluated correctly.

**Comprehensive Evaluation** This dual-framework setup enables systematic analysis across multiple dimensions: cognitive complexity (Bloom's levels), subject popularity, language (English vs. native), and modality (text vs. face image). Together, these protocols provide a rigorous, multi-dimensional evaluation of biographical reasoning in LLMs and MLLMs.

## 4 RESULTS

### 4.1 OVERALL TRENDS ACROSS COGNITIVE LEVELS

Across all benchmarks, accuracy varies systematically with the cognitive demand of the task (Table 2). Lower-order levels such as *Remembering* and *Understanding* exhibit the highest accuracies across models, while higher-order levels such as *Evaluating* and *Creating* expose more pronounced weaknesses. Closed source models (Gemini Flash 2.5, GPT-4) consistently achieve accuracies above 85–95% in most conditions, while open-source baselines (Gemma3-12b-it, Qwen2.5-7b) lag significantly, often below 60% without retrieval augmentation. This gap widens at higher cognitive levels, where reasoning demands exceed memorization.

### 4.2 MODEL COMPARISONS

**Closed-source vs. open-source.** Closed-source models clearly dominate in absolute accuracy. Gemini Flash 2.5 is the most consistent, frequently surpassing 95% with retrieval, and maintaining strong scores even without it. GPT-4 trails slightly but remains highly competitive, especially in higher-order reasoning. By contrast, Gemma3-12b-it and Qwen2.5-7b perform poorly in the zero-shot condition, particularly on less popular individuals, where accuracies often remain below 40%. However, when retrieval is introduced, both open-source models experience dramatic gains, narrowing the performance gap with closed-source systems.

**Head-to-head differences.** Between closed-source leaders, Gemini Flash 2.5 demonstrates superior factual recall (*Remembering*, *Understanding*), while GPT-4 shows more balanced performance

in higher-order reasoning (*Evaluating*, *Creating*), especially under multimodal input. Among open-source systems, Gemma3-12b-it im most cases outperforms Qwen2.5-7b, reflecting advantages of model scale and training quality. Nevertheless, both remain heavily dependent on retrieval to reach competitive levels.

### 4.3 EFFECT OF RETRIEVAL-AUGMENTED GENERATION (RAG)

The AdamRAG retrieval pipeline emerges as a key performance equalizer. For closed-source models, RAG improves factual recall and stabilizes accuracy across languages, pushing most scores above 95%. For open-source models, the gains are transformative: Qwen2.5-7b improves from sub-40% to well above 70% on mid- and high-popularity individuals, while Gemma3-12b-it achieves accuracies above 80% in *Applying* and *Analyzing*. These improvements demonstrate that retrieval substantially mitigates knowledge gaps in smaller models, reducing the reliance on large-scale pre-training alone.

### 4.4 POPULARITY EFFECTS

Popularity, denoted by the number of stars, has a strong and consistent impact. Accuracy systematically increases from $\star$ (less popular) to $\star\star\star$ (highly popular) individuals. For example, GPT-4 in zero-shot mode improves from ∼65% on low-popularity individuals to above 90% on highly popular ones. Gemini Flash 2.5 exhibits similar gains, with improvements of 15–20 points across *Remembering* and *Applying*. Open-source systems are the most sensitive: Qwen2.5-7b struggles at $\star$ (barely exceeding 20%) but reaches 50–60% at $\star\star\star$. These patterns highlight pretraining exposure as a determinant of performance, raising concerns about fairness for less-documented individuals. Retrieval alleviates, but does not fully eliminate, this disparity.

### 4.5 LANGUAGE EFFECTS

Comparisons between English (En) and Original language (Org, determined by city of birth) reveal modest but consistent advantages for Org in the zero-shot setting, especially for closed-source systems in *Remembering* and *Understanding*. This suggests cultural and linguistic grounding enhances factual recall. With retrieval enabled, however, the difference between En and Org largely vanishes, indicating that external retrieval compensates for linguistic coverage gaps in the pretrained model.

### 4.6 MULTIMODALITY AND IMAGE INPUT

The inclusion of facial image input yields mixed outcomes. Gemini Flash 2.5 maintains high performance with or without image conditioning, suggesting robustness in multimodal integration. GPT-4, however, shows a decline in *Remembering* tasks when using images without retrieval, indicating that multimodal signals can introduce noise if not paired with external evidence. Open-source models benefit modestly from image input, but the effect is inconsistent and overshadowed by the much larger impact of retrieval augmentation.

## 5 DISCUSSION AND CONCLUSIONS

In this work, we presented **ADAM**, the first framework to systematically evaluate multimodal large language models in the domain of biography. By introducing **AdamDB** and **AdamBench**, we created a large-scale, multilingual, and cognitively structured resource for probing models across six levels of Bloom's taxonomy. We further proposed **AdamRAG**, a retrieval-augmented generation system tailored for biographical reasoning, and examined its impact across models, languages, and modalities.

Our findings highlighted three central insights. First, **model scale and provenance mattered**: closed-source models such as Gemini Flash 2.5 and GPT-4 consistently outperformed open-source baselines, though retrieval augmentation narrowed the gap substantially. Second, **popularity bias proved pervasive**: all models performed significantly better on widely known individuals than on less popular ones, underscoring heavy reliance on pretraining exposure and raising concerns for fairness and inclusivity in biographical knowledge access. Third, **retrieval emerged as a crucial**

**equalizer**: AdamRAG consistently boosted performance across the board, with particularly dramatic gains for smaller open-source models, while also mitigating disparities across languages and popularity levels.

We also observed that multimodal conditioning with face images, while offering complementary context, yielded smaller and less consistent improvements compared to retrieval, especially in lower-order tasks where visual cues added limited value. Higher-order reasoning tasks such as *Evaluating* and *Creating* remained the most challenging, indicating persistent abstraction gaps even in state-of-the-art models.

Taken together, these results demonstrated that retrieval pipelines are essential for bridging the gap between open- and closed-source models, that fairness concerns must be addressed to avoid systematic underperformance on lesser-known individuals, and that multimodal grounding requires more principled integration. Future work should refine multimodal fusion strategies, design fairness-aware evaluation protocols, and extend biographical reasoning to less-documented populations, supporting the development of more accurate, culturally sensitive, and hallucination-resistant MLLMs.

## DATA AVAILABILITY

With the publication of this work, the full biographical dataset constructed for the ADAM framework, including **AdamDB** and the benchmark **AdamBench**, will be released on Hugging Face for public access and reuse. This release will include both the structured multilingual biographical records and the cognitively stratified benchmark questions, enabling reproducibility and further research on biographical reasoning in language models.

For the review process, we provide a representative sample of the dataset in the supplementary material. This sample includes a subset of records and example benchmark questions, illustrating the data schema and evaluation design without requiring access to the full release.

## ETHICS STATEMENT

This work involved no experiments with human subjects or sensitive personal data. All biographical information was derived from publicly available sources, primarily Wikipedia and Wikidata, and processed in compliance with their respective licenses. For multimodal content, we did not redistribute copyrighted images directly. Instead, we preserved copyright by including only public URLs that reference the original hosting websites, ensuring that attribution and usage rights remain intact.

In preparing this manuscript, we employed large language models as auxiliary tools for two purposes: **(i)** text polishing, to improve clarity and readability, and **(ii)** retrieval and discovery, to aid in literature review (e.g., via tools such as *ScholarQA*[1]). All scientific content, methodological design, and experimental results were conceived, executed, and validated by the authors. The use of AI tools was limited to supportive roles, and we disclose this practice for transparency in line with emerging community standards.

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

## A    ADDITIONAL DATASET STATISTICS

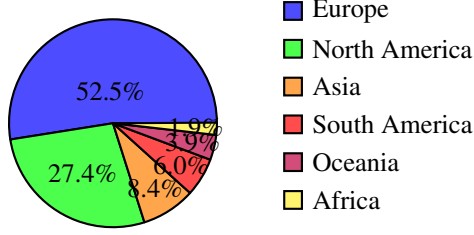

Figure 2: Distribution of individuals by continent in AdamDB.

Table 3: Language coverage statistics.

| Language | Names | Coverage % |
|---|---|---|
| English (en) | 3,973,119 | 99.7 |
| Dutch (nl) | 3,423,044 | 85.9 |
| Spanish (es) | 2,593,617 | 65.1 |
| French (fr) | 2,070,396 | 52.0 |
| German (de) | 1,921,806 | 48.2 |
| Italian (it) | 1,580,213 | 39.7 |
| Portuguese (pt) | 1,225,397 | 30.8 |
| Polish (pl) | 649,224 | 16.3 |
| **Major 8 Languages** | **17,436,816** | **Avg 55.9%** |

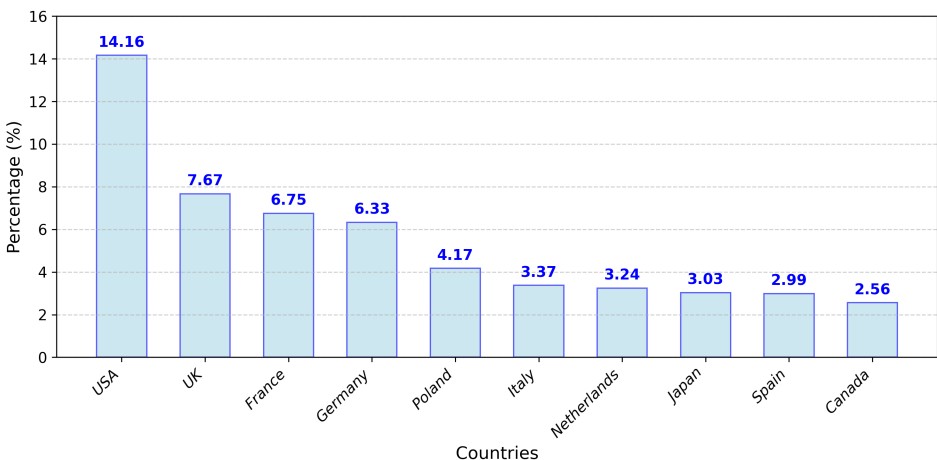

Figure 3: Distribution of top 10 countries by records in AdamDB.

