# OpenReview forum: "ADAM: A Diverse Archive of Mankind for Multimodal Benchmarking and Enhancing LLMs’ Cognitive Skills in Biographical Contexts"
_ICLR.cc/2026/Conference — ICLR 2026 Conference Withdrawn Submission_

### Official Review · Reviewer_hrrh · 2025-10-20

**Soundness:** 2
**Presentation:** 2
**Contribution:** 2
**Rating:** 2
**Confidence:** 2

**Summary:**

This paper introduces ADAM, a comprehensive framework designed to evaluate and enhance the biographical reasoning capabilities of Large Language Models (LLMs). The framework consists of three main components: (1) AdamDB, a large-scale, multilingual, and multimodal knowledge base covering over 4 million individuals; (2) AdamBench, an evaluation benchmark structured around Bloom's Taxonomy to assess six levels of cognitive reasoning ; and (3) AdamRAG, a Retrieval-Augmented Generation (RAG) system that leverages AdamDB to improve factual accuracy and reduce hallucinations. Through extensive experiments, the authors demonstrate that their RAG system significantly improves performance, particularly for open-source models and lesser-known individuals, while also highlighting the pervasive issue of popularity bias in current LLMs.

**Strengths:**

The paper's primary strength is its framework that addresses biographical reasoning through data creation, a novel evaluation benchmark, and system improvement. It contributes AdamBench, a valuable benchmark grounded in Bloom's Taxonomy that enables a deeper assessment of cognitive reasoning beyond simple factual recall . Furthermore, the study provides a significant large-scale resource in AdamDB and offers strong empirical evidence for the transformative impact of RAG and the pervasive problem of popularity bias in current LLMs .

**Weaknesses:**

Unclear Motivation: The paper fails to clearly articulate whether its primary goal is to improve LLM applications for biography or to simply use biography as a scenario for studying LLM capabilities.

Vague Methodology: The construction of AdamDB, the question generation for AdamBench, and the retrieval pipeline for AdamRAG all lack critical technical details, specific examples, and parameter settings.

Fundamentally Flawed Benchmark:

Questionable Generation Process: The benchmark's reliance on an LLM for question generation is undermined by the complete absence of prompt examples and any description of a human quality assurance process, calling its scientific validity into question.

Method-Objective Mismatch: The multiple-choice question format is methodologically incapable of properly evaluating high-level, open-ended cognitive skills like "Creating."

Severely Biased Core Definitions: Defining "popularity" by English Wikipedia page views introduces a strong Anglo-centric and recency bias; determining "native language" by "city of birth" is overly simplistic and often inaccurate.

Confusing and Unintuitive Presentation of Results: The core results in Table 2 are poorly designed and difficult to read, and the paper completely lacks data visualization, which weakens the impact of its findings.

**Questions:**

Can you provide the specific prompts used to generate the AdamBench questions, particularly the instructions for guiding the LLM to create questions for different cognitive levels?

Was there a human review or quality assurance process during the construction of AdamBench? If so, what was the procedure? If not, how can the benchmark's validity be guaranteed?

How do you justify that a multiple-choice question format can effectively and accurately evaluate the highest-order cognitive skill, "Creating"?

Why was English Wikipedia page views chosen as the sole metric for measuring the popularity of global figures? How did you assess and mitigate the inherent Anglo-centric bias of this metric?

In the AdamRAG system, what is the mechanism for handling cases where information is not found or where conflicting information is retrieved? Why were no ablation studies conducted to justify the necessity of each component in your retrieval pipeline?

How does your framework ensure that it does not become a "bias amplifier" by rewarding models that perform better on well-known figures from the Western world, thereby exacerbating the neglect of figures from other cultures?

---

### Official Review · Reviewer_75Dr · 2025-10-26

**Soundness:** 3
**Presentation:** 2
**Contribution:** 3
**Rating:** 4
**Confidence:** 3

**Summary:**

The paper introduces ADAM, an end-to-end framework for biographical reasoning that couples a massive multilingual/multimodal knowledge base (AdamDB, 4M+ people, around 600 languages) with a cognitively structured benchmark (AdamBench, Bloom’s six levels) and a tailored retrieval system (AdamRAG). Experiments show AdamRAG substantially improves open-source models and modestly helps closed-source ones, with the largest gains on lower-order cognition; popularity (celebrity vs. long-tail) strongly affects accuracy, and face images add smaller, less consistent gains than retrieval. ADAM provides a first comprehensive, popularity-aware, multilingual, multimodal evaluation pipeline for biography and highlights fairness and grounding gaps in current LLMs.

**Strengths:**

1. The paper proposes a valuable data resource (AdamDB): a large-scale, multilingual, and multimodal biographical database that meaningfully extends coverage to long-tail and low-resource entities, providing a solid foundation for reliable retrieval-based reasoning.
2. The paper introduces a well-designed benchmark (AdamBench).

**Weaknesses:**

1.  Experimental Analysis and Ablations.The experimental section lacks sufficient ablation to isolate the effect of individual components in AdamRAG. It remains unclear which design choices (e.g., nationality filters, temporal constraints, embedding selection) drive the observed performance gains. Future work should include controlled studies to clarify these contributions and the robustness of retrieval under noisy or incomplete data.
2.  About Multimodal Component. The exploration of multimodality is limited. Although face images are included, their contribution appears minor or inconsistent across models, and the paper provides little analysis of why. Are these results due to weak visual-text alignment, low-quality images, or insufficient fusion mechanisms? A deeper exploration of visual reasoning could enhance the multimodal claim.
3.  Visualization and Qualitative Results. The results are purely quantitative; no qualitative examples or case studies. Showing concrete examples, such as retrieved evidence, reasoning chains, or success/failure cases would make the findings more interpretable and persuasive.
4.  Benchmark Challenge Level. The benchmark may be overly solvable for top proprietary models, with accuracies approaching 90–95% on many tasks. This raises questions about whether AdamBench can continue to challenge next-generation LLMs. The authors could consider incorporating harder, open-ended reasoning or causal inference tasks to better reflect real-world difficulty and maintain future relevance.

**Questions:**

See weaknesses.

---

### Official Review · Reviewer_DtkC · 2025-10-31

**Soundness:** 3
**Presentation:** 3
**Contribution:** 3
**Rating:** 6
**Confidence:** 2

**Summary:**

The paper introduces ADAM, a comprehensive, multilingual, multi-regional biographical database (adamDB), benchmark (adamBench), and a RAG system (adamRAG) for improving and evaluating LLMs' understanding and knowledge about biographical information. With the proposed system, the paper hope to identify, and reduce the hallucination of LLMs when it comes to biographical information understanding and usage. Also, by evaluating LLMs on the proposed benchmark, it is able to expose certain bias and trend in the current LLMs that are useful for study.

**Strengths:**

1. A biographical database and its associated benchmark+RAG system are a necessary and important contribution to the LLM benchmark community, as biographical queries can potentially account for a large portion of users' demand in their daily lives when using LLMs as a search engine. Therefore, having a system that strives to evaluate and improve this front could be valuable to the field.
2. The paper proposed a complete system that includes both improving the LLMs and testing them on biographical questions. It seems that others can directly use the RAD system and benchmark for their own purpose without much modification and additional effort.
3. The database proposed is multilingual and covers a vast number of countries and demographics.

**Weaknesses:**

1. The title includes the word Mankind, which could be potentially misleading. I'd suggest changing it to a gender-neutral word, such as humankind or others.
2. It seems that the models already do a decent job at some of the tasks listed in Table 2. So it makes me question the value of the proposed RAG system. Further, I'm not sure if the tested LLMs have access to the internet. I would imagine that for the newer version of the models, which can automatically search the web, their performance would be even higher.

**Questions:**

1. It would be great to see more breakdown of the database. For example, gender, era, etc..

---

### Official Review · Reviewer_zrz7 · 2025-11-03

**Soundness:** 2
**Presentation:** 3
**Contribution:** 2
**Rating:** 4
**Confidence:** 3

**Summary:**

I have reviewed this paper which presents ADAM (A Diverse Archive of Mankind), a comprehensive framework for evaluating and enhancing multimodal large language models (MLLMs) in biographical reasoning tasks. The framework consists of three main components: (1) AdamDB, a large-scale multilingual biographical knowledge base containing over 4 million individuals across 595 languages; (2) AdamBench, an evaluation benchmark structured around Bloom's Taxonomy covering six cognitive levels from remembering to creating; and (3) AdamRAG, a customized retrieval-augmented generation system incorporating cross-lingual retrieval and popularity-weighted search. The authors conduct experiments demonstrating that AdamRAG improves accuracy over zero-shot baselines, particularly for open-source models and less popular individuals, and reveal systematic popularity bias in current MLLMs.

**Strengths:**

**[+] Substantial dataset contribution:** The AdamDB dataset represents a significant resource contribution to the community. With over 4 million biographical records covering 595 languages and 200+ countries, this substantially expands the available resources for biographical reasoning research. The data construction pipeline, from WikiDB extraction through NER filtering and multilingual alignment, appears methodologically sound and reproducible.

**[+] Multi-dimensional evaluation design:** I appreciate the paper's attempt to evaluate MLLMs across multiple important dimensions simultaneously: cognitive complexity (via Bloom's Taxonomy), popularity bias, multilingual capability, and multimodal reasoning. This multi-faceted perspective provides richer insights than single-metric evaluations.

**[+] Important findings on popularity bias:** The experiments clearly demonstrate systematic popularity bias in current MLLMs, with accuracy rates varying dramatically between high-popularity and low-popularity individuals (e.g., 53.7% vs 19.7% for Qwen2.5-7b). This is an important finding that highlights limitations in how these models handle long-tail knowledge.

**[+] Clear presentation and organization:** The paper is generally well-written with clear motivation, and Figure 1 effectively communicates the overall framework architecture.

**Weaknesses:**

**[-] Limited experimental validation of AdamRAG:** My primary concern is that the experimental evaluation only compares AdamRAG against zero-shot baselines. While this demonstrates that retrieval helps (which is expected), it does not validate whether AdamRAG's specific design choices (popularity-weighted search, cross-lingual retrieval) provide advantages over standard RAG implementations. I strongly encourage the authors to include comparisons with at least a basic dense retrieval baseline and ideally one or two established RAG systems. This would substantially strengthen the claims about AdamRAG's effectiveness.

**[-] Related work coverage could be more comprehensive:** I noticed that some relevant work on RAG systems and factuality evaluation could be better covered. For instance, Self-RAG (Asai et al., ICLR 2024) represents an important advance in RAG systems designed to reduce hallucinations, which seems directly relevant to this work's goals. Additionally, the foundational RAG paper (Lewis et al., 2020) should be cited. More comprehensive coverage would help readers better understand how this work fits into the broader landscape and would strengthen the paper's positioning.

**[-] Novelty claims could be more precisely stated:** The paper claims to be the "first" to systematically evaluate biographical reasoning using Bloom's Taxonomy and to study long-tail knowledge effects. However, I would suggest being more careful with such claims, as similar evaluation frameworks have been explored in recent literature. I recommend framing the contribution more specifically as applying these evaluation approaches to the biographical domain with a uniquely large-scale multilingual dataset, which is still a valuable contribution without requiring claims of absolute novelty.

**Questions:**

**Q1:** Could you provide experimental comparisons between AdamRAG and at least one standard RAG baseline (e.g., a basic dense retrieval approach using a standard encoder)? This would help clarify whether the specific design choices in AdamRAG (particularly popularity-weighted search) provide meaningful advantages.

**Q2:** Have you considered evaluating AdamRAG on an independent, established benchmark (such as existing factuality or knowledge-intensive QA benchmarks) to demonstrate its generalization capability beyond AdamBench?

**Q3:** Could you elaborate on the computational cost comparison between AdamRAG and standard RAG approaches? Given the additional components (popularity weighting, cross-lingual retrieval), it would be helpful to understand the efficiency trade-offs.

**Q4:** For the popularity-weighted search component, how sensitive is performance to the choice of weighting parameters? Did you conduct ablation studies to validate this design choice?

**Q5:** In the related work section, could you discuss how AdamBench differs from or complements other recent efforts to evaluate LLMs on factuality and long-tail knowledge? This would help clarify the specific niche that ADAM fills.

---

### Note · Authors · 2025-12-20

I have read and agree with the venue's withdrawal policy on behalf of myself and my co-authors.